# Cost-effectiveness of leveraging existing HIV primary health systems and community health workers for hypertension screening and treatment in Africa: An individual-based modeling study

Matthew D. Hickey[1]*, James Ayieko[2], Jane Kabami[3], Asiphas Owaraganise[3], Elijah Kakande[3], Sabina Ogachi[2], Colette I. Aoko[2], Erick M. Wafula[2], Norton Sang[2], Helen Sunday[3], Paul Revill[4], Loveleen Bansi-Matharu[5], Starley B. Shade[6], Gabriel Chamie[1], Laura B. Balzer[7], Maya L. Petersen[7], Diane V. Havlir[1], Moses R. Kamya[3,8‡], Andrew N. Phillips[5‡]

1 Division of HIV, Infectious Diseases, and Global Medicine, University of California San Francisco, San Francisco, California, United States of America, 2 Kenya Medical Research Institute, Nairobi, Kenya, 3 Infectious Diseases Research Collaboration, Kampala, Uganda, 4 Centre for Health Economics, University of York, York, United Kingdom, 5 Institute for Global Health, University College London, London, United Kingdom, 6 Department of Epidemiology and Biostatistics, University of California San Francisco, San Francisco, California, United States of America, 7 School of Public Health, University of California, Berkeley, Berkeley, California, United States of America, 8 Department of Medicine, Makerere University, Kampala, Uganda

‡ These authors are co-senior authors on this work.
* matt.hickey@ucsf.edu

## Abstract

### Background

Cardiovascular disease (CVD) morbidity and mortality is increasing in Africa, largely due to undiagnosed and untreated hypertension. Approaches that leverage existing primary health systems could improve hypertension treatment and reduce CVD, but cost-effectiveness is unknown. We evaluated the cost-effectiveness of population-level hypertension screening and implementation of chronic care clinics across eastern, southern, central, and western Africa.

### Methods and findings

We conducted a modeling study to simulate hypertension and CVD across 3,000 scenarios representing a range of settings across eastern, southern, central, and western Africa. We evaluated 2 policies compared to current hypertension treatment: (1) expansion of HIV primary care clinics into chronic care clinics that provide hypertension treatment for all persons regardless of HIV status (chronic care clinic or CCC policy); and (2) CCC plus population-level hypertension screening of adults ≥40 years of age by community health workers (CHW policy). For our primary analysis, we used a cost-effectiveness threshold of US $500 per disability-adjusted life-year (DALY) averted, a 3% annual discount rate, and a 50-year

**Data Availability Statement:** The model program and dataset are available from the Dryad database: https://doi.org/10.5061/dryad.5x69p8d9b

**Funding:** This work was supported by the National Heart, Lung, and Blood Institute (K23HL162578 to MDH) and the National Institute of Allergy and Infectious Diseases, the National Heart, Lung, and Blood Institute, the National Institute of Mental Health, and the National Institute on Alcohol Abuse and Alcoholism (U01-AI150510 to DVH, MLP, and MRK). The funders had no role in study design, data collection and analysis, decision to publish, or preparation of the manuscript.

**Competing interests:** The authors have declared that no competing interests exist.

**Abbreviations:** CCC, chronic care clinic; CHW, community health worker; CVD, cardiovascular disease; DALY, disability-adjusted life-year; GDP, gross domestic product; ICER, incremental cost-effectiveness ratio; IHD, ischemic heart disease; LMIC, low and middle-income country; SBP, systolic blood pressure; SEARCH, Sustainable East Africa Research in Community Health; SOC, standard of care.

time horizon. A strategy was considered cost-effective if it led to the lowest net DALYs, which is a measure of DALY burden that takes account of the DALY implications of the cost for a given cost-effectiveness threshold.

Among adults 45 to 64 years, CCC implementation would improve population-level hypertension control (the proportion of people with hypertension whose blood pressure is controlled) from mean 4% (90% range 1% to 7%) to 14% (6% to 26%); additional CHW screening would improve control to 44% (35% to 54%). Among all adults, CCC implementation would reduce ischemic heart disease (IHD) incidence by 10% (3% to 17%), strokes by 13% (5% to 23%), and CVD mortality by 9% (3% to 15%). CCC plus CHW screening would reduce IHD by 28% (19% to 36%), strokes by 36% (25% to 47%), and CVD mortality by 25% (17% to 34%). CHW screening was cost-effective in 62% of scenarios, CCC in 31%, and neither policy was cost-effective in 7% of scenarios. Pooling across setting-scenarios, incremental cost-effectiveness ratios were $69/DALY averted for CCC and $389/DALY averted adding CHW screening to CCC.

## Conclusions

Leveraging existing healthcare infrastructure to implement population-level hypertension screening by CHWs and hypertension treatment through integrated chronic care clinics is expected to reduce CVD morbidity and mortality and is likely to be cost-effective in most settings across Africa.

## Author summary

### Why was this study done?

- Cardiovascular disease (CVD), such as heart attacks and strokes, are increasingly causing illness and death in Africa, mainly due to undiagnosed and untreated hypertension.

- Hypertension is currently treated primarily in specialized clinics; existing primary health systems, particularly those developed for HIV care, could potentially be used to treat hypertension more effectively.

- Prior research also demonstrates that community health workers can successfully conduct hypertension screening in the community, improving both diagnosis and linkage to care.

- This research aimed to determine whether integration of hypertension care within existing primary health systems with or without community health worker screening of all adults aged 40 or greater in the community for hypertension would be a worthwhile investment in Africa.

## What did the researchers do and find?

- We incorporated hypertension and CVD into an existing individual-level HIV model and simulated hypertension and CVD outcomes across 3,000 scenarios in eastern, southern, central, and western Africa.

- We evaluated 2 policies compared to current standard hypertension care: expanding primary care clinics to include HIV and hypertension (chronic care clinic or CCC policy) and adding community health worker (CHW) screening for all adults over 40 years of age.

- For all adults, CCC would reduce heart attacks by 10%, strokes by 13%, and cardiovascular deaths by 9%, while adding CHW screening would reduce these by 28%, 36%, and 25%, respectively.

- CHW screening was cost-effective in 62% of scenarios, CCC in 31%, and neither policy was considered cost-effective in 7% of scenarios.

## What do these findings mean?

- Using existing primary health care and community health worker infrastructure to screen and treat hypertension can significantly reduce illness and death from CVD in Africa.

- Implementing chronic care clinics and community health worker screenings for hypertension is likely to be cost-effective in most settings.

- These findings support policy changes to integrate hypertension management into existing primary health services to improve CVD prevention.

- Though we considered numerous factors in our model, findings are limited by uncertainty in model parameters, inability to include all potential policy alternatives, and uncertainty around how the current state of hypertension care will evolve in the future.

## Introduction

Hypertension is the leading preventable cause of morbidity and premature mortality globally [1] and the main driver of the rising burden of cardiovascular disease (CVD) in Africa [2]. Hypertension can be controlled with highly efficacious [3], low-cost [4] medications; however, current implementation of hypertension treatment is limited and often only available in centralized hypertension clinics that are inaccessible to many people due to long transportation times, high out-of-pocket costs, and competing priorities [5]. Under current conditions, as low as one-quarter of persons with hypertension in eastern, southern, central, and western Africa have been diagnosed, and fewer than 10% have controlled hypertension [6–9].

One approach to address the large unmet need for hypertension care would be to leverage existing chronic care systems. Initially developed as an emergency response to the HIV pandemic, HIV primary care clinics now provide long-term HIV care to more than 25 million people with HIV at decentralized primary health centers across Africa [10]. As such, HIV

clinics are well-positioned to deliver hypertension and other chronic disease care that is close to where people live, facilitating ease of access and continuity of care [11,12]. Several large randomized trials provide evidence for implementation of hypertension care that is integrated within HIV primary care clinics for people with and without HIV. The Sustainable East Africa Research in Community Health (SEARCH) cluster randomized trial in rural Kenya and Uganda tested a population-based multi-disease approach for HIV and hypertension screening and treatment, including implementation of hypertension care in government-run HIV primary care clinics [13]. This integrated approach to hypertension care improved retention in care, hypertension control, and all-cause mortality [14]. The INTE-AFRICA study in Uganda and Tanzania similarly showed that chronic care clinics providing hypertension, diabetes, and/or HIV care improved retention in care and adherence while reducing duplication of services and costs, compared to non-integrated care [15,16].

Integrating hypertension treatment within primary health centers may improve treatment outcomes among those already accessing clinic-based services; however, such interventions may have limited population-level impact without broader hypertension screening interventions to address low levels of diagnosis [7]. Community-based hypertension screening by existing community health workers (CHWs) is one strategy to improve hypertension awareness at a population level. CHWs are paid or volunteer lay community members who operate as a bridge between primary health systems and the community. CHWs in low and middle-income countries (LMICs) can be trained to correctly perform community-based hypertension screening [17,18] and can successfully identify persons with undiagnosed hypertension through community-based screening, link them to care, and improve blood pressure control [19–22]. CHWs are present in many settings in Africa, including both rural and urban settings, and may represent a broadly generalizable strategy for increasing hypertension awareness [23,24].

We conducted a modeling study to evaluate the effectiveness and cost-effectiveness of scaling up 2 different policies for hypertension screening and treatment across settings in Africa, compared to current hypertension treatment conditions. The policies we consider include (1) expansion of HIV primary care clinics into chronic care clinics that provide hypertension care for all persons regardless of HIV status; and (2) chronic care clinic implementation plus population-level hypertension screening of adults aged ≥40 years by CHWs.

## Methods

### Model description

Detailed methods and model parameters are provided in S1 Appendix (Sections 1–3, pages 3–24); here, we provide a summary. We incorporated systolic blood pressure (SBP) and how it changes over the life course into the HIV Synthesis model, an individual-based simulation model of HIV transmission, treatment, morbidity, and mortality that has been described previously [25]. Each run of the model generates a simulated population beginning in 1989. Individual variables updated in 3-monthly periods include age, sex, true underlying SBP, presence of a hypertension clinic visit, measured SBP (if measured), hypertension diagnosis (defined as measured SBP ≥140 mmHg on 2 consecutive occasions, ≥160 mmHg on one occasion, or initiation of antihypertensive therapy), initiation of antihypertensive therapy, number of antihypertensive drugs, incident ischemic heart disease (IHD), incident stroke, and death. The underlying HIV Synthesis model also includes HIV-related variables including time-updated HIV status, CD4 count, HIV viral load, antiretroviral therapy treatment, HIV clinic visits, and HIV-associated morbidity and mortality. We modeled incidence of CVD events and the dependence on SBP, sex, age, HIV status, and prior history of CVD.

For each individual in the model at each 3-month time step, we explicitly considered whether blood pressure was measured, the measured value (incorporating measurement variability), as well as diagnosis and treatment of hypertension. The model incorporates both patient- and provider-level behavior regarding uptake of hypertension screening, diagnosis, treatment, and retention, including that some individuals will not uptake any health interventions. We summarize in Fig 1 our overall approach to modeling hypertension screening, diagnosis, treatment initiation, and retention in care—and how treatment-related reductions in SBP influence risk for incident CVD. As an individual-based model, variables described in Fig 1 are updated for each person in the model at every 3-month time step. The full individual model is defined by the set of variables and the updating rules which can be seen in full in S1 Appendix (Section 3, pages 6–24).

We generated 3,000 setting-scenarios to represent a range of settings across eastern, southern, central, and western Africa and to incorporate uncertainty into model outputs. A setting-scenario refers to a single run of the model where values for each model parameter are independently and randomly sampled from the set of parameter values. The combined set of model parameters is intended to represent a particular setting (national or subnational) across settings in eastern, southern, central, and western Africa, as well as uncertainty around parameter values.

We calibrated model estimates for hypertension prevalence, diagnosis, treatment and blood pressure control under standard of care conditions to available data from the literature (Table 1; Section 4, pages 25–35 of S1 Appendix). In 2015, the mean (90% range) model-estimated hypertension prevalence among adults aged 45 to 64 was 42% (30% to 54%). An estimated 28% (18%

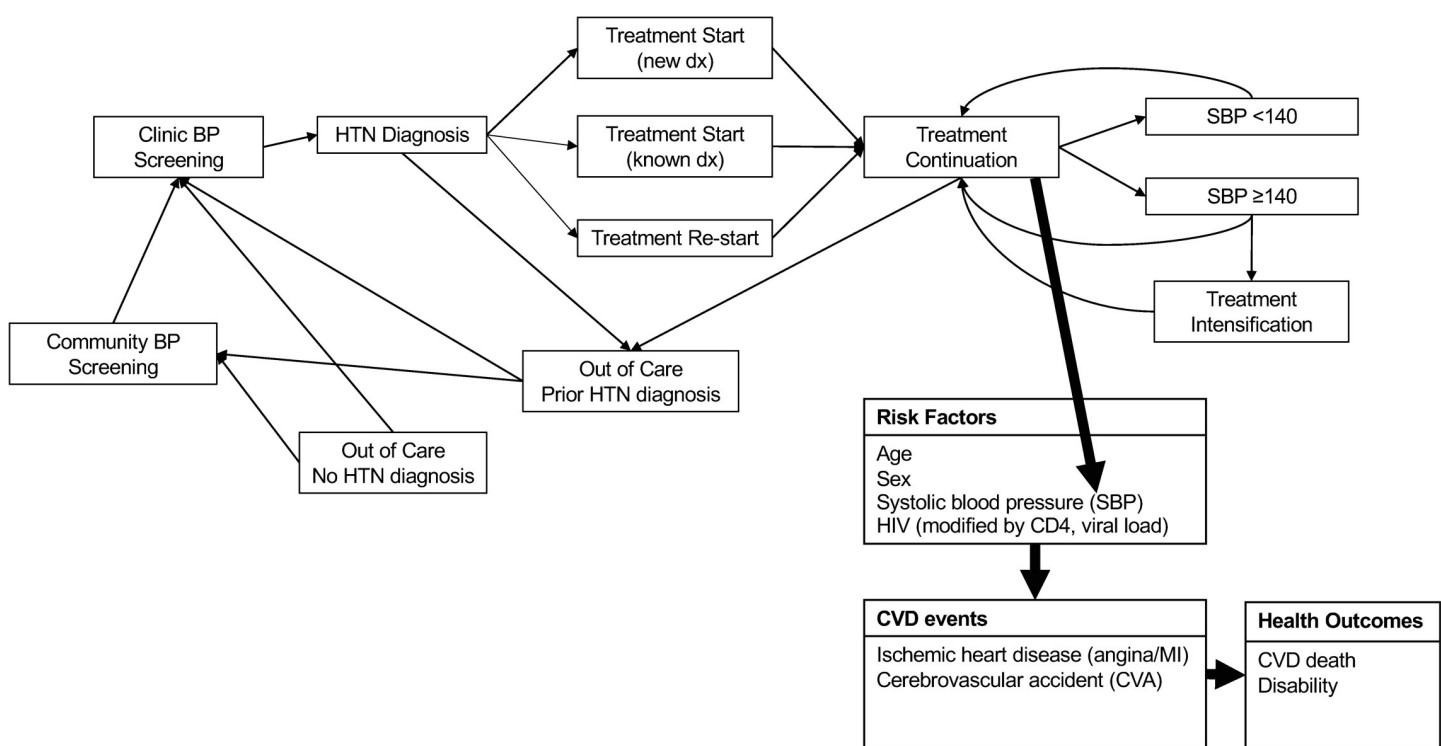

**Fig 1. Simplified model schematic.** Simplified model schematic to provide a broad overview of how individual persons in the model move through the continuum of hypertension care over time and how hypertension treatment impacts CVD outcomes. This is an individual model where variables for each individual are updated each 3-month time step. The full individual model is defined by the set of variables and the updating rules which can be seen in full in S1 Appendix. CVD, cardiovascular disease; SBP, systolic blood pressure.

**Table 1. Characteristics relating to hypertension in 3,000 setting scenarios in 2015\*.**

| | Age category: Mean (90% range) | Examples of observed data [7] |
|---|---|---|
| *Hypertension prevalence* | 25–34 18% (13% to 22%)<br>35–44 26% (17% to 35%)<br>45–54 38% (26% to 50%)<br>55–64 48% (37% to 60%)<br>≥65 58% (47% to 67%) | 25–34: Burkina Faso 2013 11%, eSwatini 2014 16%, Kenya 2015 14%, South Africa 2012 22%, Tanzania 2012 16%, Togo 2010 15%, Uganda 2014 20%<br>35–44: Burkina Faso 2013 17%, eSwatini 2014 30%, Kenya 2015 30%, South Africa 2012 38%, Tanzania 2012 24%, Togo 2010 25%, Uganda 2014 24%<br>45–54: Burkina Faso 2013 27%, eSwatini 2014 43%, Kenya 2015 41%, South Africa 2012 58%, Tanzania 2012 41%, Togo 2010 38%, Uganda 2014 39%<br>55–64: Burkina Faso 2013 39%, eSwatini 2014 66%, Kenya 2015 49%, South Africa 2012 68%, Tanzania 2012 52%, Togo 2010 47%, Uganda 2014 51%<br>≥65: eSwatini 2014 66%, Kenya 2015 57%, South Africa 2012 85%, Uganda 2014 51% |
| *Proportion of hypertensive people diagnosed* | 25–34 10% (6% to 16%)<br>35–44 16% (9% to 24%)<br>45–54 24% (15% to 36%)<br>55–64 31% (20% to 44%)<br>≥65 37% (25% to 52%) | 25–34: Burkina Faso 2013 13%, eSwatini 2014 21%, Kenya 2015 13%, South Africa 2012 22%, Tanzania 2012 10%, Togo 2010 8%, Uganda 2014 10%<br>35–44: Burkina Faso 2013 10%, eSwatini 2014 38%, Kenya 2015 20%, South Africa 2012 27%, Tanzania 2012 20%, Togo 2010 22%, Uganda 2014 18%<br>45–54: Burkina Faso 2013 24%, eSwatini 2014 50%, Kenya 2015 37%, South Africa 2012 44%, Tanzania 2012 21%, Togo 2010 30%, Uganda 2014 32%<br>55–64: Burkina Faso 2013 34%, eSwatini 2014 62%, Kenya 2015 34%, South Africa 2012 47%, Tanzania 2012 20%, Togo 2010 37%, Uganda 2014 20%<br>≥65: eSwatini 2014 59%, Kenya 2015 31%, South Africa 2012 52%, Uganda 2014 31% |
| *Proportion of hypertensive people on pharmacologic treatment†* | ≥18 4% (2% to 8%)<br>25–44 2% (1% to 4%)<br>45–64 6% (3% to 12%)<br>≥65 9% (3% to 16%) | Burkina Faso 2013 9%, eSwatini 2014 21%, Kenya 2015 8%, South Africa 2012 28%, Tanzania 2012 6%, Togo 2010 8%, Uganda 2014 8% |
| *Proportion of hypertensive people with controlled hypertension‡* | ≥18 2% (1% to 5%)<br>25–44 1% (0% to 3%)<br>45–64 3% (1% to 7%)<br>≥65 4% (1% to 9%) | Burkina Faso 2013 4%, eSwatini 2014 7%, Kenya 2015 3%, South Africa 2012 9%, Tanzania 2012 2%, Togo 2010 3%, Uganda 2014 2% |

\* Model estimates are averaged across 2013 through 2016. We report estimates from this time period because this approximates the time period of most currently available hypertension care cascade data from the literature.

† In the model, defined as prescribed and taking (adherent to) antihypertensive medications. In the literature, treatment defined self-report of currently taking antihypertensive medication.

‡ Defined as SBP <140 mmHg in the model and BP <140/90 mmHg in the literature.

to 40%) had ever been diagnosed, 6.3% (3% to 12%) were on treatment, and 3.4% (1% to 7.1%) had controlled hypertension. Median (90% range) model-estimated HIV prevalence was 11% (2% to 34%) among adults aged 15 to 49 years across setting-scenarios in 2015.

## Policy comparison

For each setting-scenario, we considered the situation at the end of 2023 and compared 2 policies to maintenance of current hypertension care (standard of care; SOC) (Table 2): (1)

**Table 2. Hypertension policies considered.**

|  | Standard care (SOC) | Chronic care clinic (CCC) | Chronic care clinic + CHW screening (CHW) |
|---|---|---|---|
| Community-based screening | none | none | Annual CHW screening for adults ≥40* |
| Treatment | Standard care in hypertension clinic | Hypertension treatment integrated within HIV primary care clinic system | Hypertension treatment integrated within HIV primary care clinic system |

* We conducted sensitivity analysis evaluating the CHW policy with screening limited to adults ≥50 years of age.

CCC, chronic care clinic; CHW, community health worker; SOC, standard of care.

expansion of HIV primary care clinics into chronic care clinics that provide integrated hypertension care for all persons regardless of HIV status (chronic care clinic or CCC policy); and (2) CCC implementation plus annual population-level hypertension screening of adults aged ≥40 years by CHWs (CHW policy). We also conducted sensitivity analysis of the CHW policy evaluating screening limited to adults ≥50 years.

The SOC policy represents the current state of hypertension care delivery, which is often only available in centralized hospital-based hypertension clinics that are inaccessible to many people due to long transportation times, high out-of-pocket costs, and competing priorities [5]. The CCC policy entails integration of hypertension screening and treatment within primary care clinics, as conceptualized in both the SEARCH and INTE-AFRICA studies and including hypertension care delivery for both people with and without HIV [14–16]. Key strategies in these studies include provider training on patient-centered care/friendly services and hypertension management, changes in clinic workflows to incorporate routine blood pressure screening and nursing triage, and adoption of clinic record systems that mirrored those used for long-term monitoring of HIV (e.g., patient cards to record blood pressure and medications over multiple visits). Such interventions were low cost but effective for improving blood pressure treatment and control compared to non-integrated care [14,26]. In both SOC and CCC policies, we presume that blood pressure screening is only conducted within clinic-based settings; we further presume that the CCC model does not increase rates of presenting to clinic for hypertension screening. The CHW policy represents a frameshift by implementing hypertension screening in the community, by CHWs. We presume that screening is conducted at the household level, alongside other CHW-based household interventions. For each of these policies, we modeled individual-level effects on hypertension screening, diagnosis, and treatment over time based on data from the SEARCH studies (Tables D–F and H, pages 6–20 of S1 Appendix) [14,21,27].

For SOC conditions, we assumed that hypertension visits were scheduled once per quarter (the smallest time step in the model), though assumed a higher visit cost during visits when blood pressure was uncontrolled (SBP ≥140) to account for longer visit time or additional visits during that quarter (see Cost effectiveness section). For CCC and CHW policies, we assume that the chronic care model includes multi-month prescribing for controlled hypertension and thus that visits are scheduled every 6 months when SBP is <140 mmHg and every 3 months when SBP is ≥140 mmHg [28]. For all modeling parameters, we included a range of values to reflect variability in effectiveness and implementation across settings (detailed intervention parameters in Table H, pages 19 and 20 of S1 Appendix).

## Cardiovascular disease

We modeled incident IHD and stroke events (collectively, CVD events) using risk equations incorporating age, SBP [29], HIV (including CD4 and HIV viral load) [30], history of CVD

[31], and setting-scenario variability in baseline CVD risk, calibrated to Global Burden of Disease estimates (Section 4, pages 25–35 of S1 Appendix) [32]. We estimate the relationship between SBP lowering down to a threshold of 115 mmHg and CVD risk based on meta-analyses of randomized controlled trials of antihypertensive medications, incorporating interaction with age [29]. For incident CVD events, we model a distribution of event severity (mild, moderate, severe) with an associated gradation in acute and long-term mortality risk (Table F, pages 12–18 of S1 Appendix) and disability weights (Table G, page 18 of S1 Appendix), modeling mortality risk using real-world data where possible [33,34]. For IHD events, consistent with Global Burden of Disesase disability weight classifications [32], we considered moderate/severe IHD events to be myocardial infarctions and mild IHD events to be incident angina or silent/minimally symptomatic myocardial infarction. When a CVD event occurs, the person experiencing the event begins accruing disability-adjusted life years (DALYs) according to the severity of the event (disability weights included in Table G, page 18 of S1 Appendix). For individuals in the model, DALYs accrue from CVD events and/or death up until 80 years of age.

We also incorporated utilization of emergency medical care for incident CVD events, assuming only those with moderate or severe events would potentially seek care and that there is setting-scenario variability in the proportion of individuals who seek care following an event (Table D, pages 6–10 of S1 Appendix). When effective emergency CVD care is accessed, we model risk reduction in acute mortality based on trial data for medical therapies that are likely to be available in most settings in Africa (e.g., aspirin, supportive care) [35,36]. We further assume that at least some care is delayed or inadequate and therefore not effective for reducing acute CVD mortality [37].

## Cost estimates

We provide assumed costs in S1 Appendix (Table I, pages 21–23). For all costs, we calculated a base-case and both 50% and 200% cost scenarios. In our base-case, we assume a cost of $3 United States dollars (USD) per person screened by a CHW (inclusive of CHW salary, supervisor salary, blood pressure cuff, annual training, and overhead/implementation costs), clinic visits of $10 USD if hypertension is uncontrolled (SBP ≥140 mmHg) and $5 USD if controlled (SBP <140 mmHg) [26,38], and 3-month drug costs of $1.5 USD for first and second line and $3 USD for third line antihypertensive therapy (assuming drugs are available at generic costs plus 30% to account for supply chain costs) [39]. CHW compensation rates were based on the median minimum wage across countries in Africa that have revised minimum wage policies within the past 5 years; sensitivity analyses encompass the range of minimum wages across settings (Tables I and J, pages 21–24 of S1 Appendix). To account for real-world implementation, our assumed visit costs are approximately double that observed in time and motion studies from the literature [26,38]. We also included a fixed annual cost of initial and ongoing annual training for clinic staff in the CCC and CHW policies to facilitate implementation of the chronic care model. Two large studies of integrated HIV and hypertension care found that hypertension visits did not add any cost when integrated with an HIV care visit, compared to HIV care visits alone (or in some cases were cost-saving) [26,38]. To address situations where integration is incomplete, we assumed 2 different costs for hypertension visits when integrated with HIV care. In half of the scenarios, we assumed no additional cost for a hypertension clinic visit when it was combined with an HIV care visit. In the other half, we assumed the cost for adding hypertension care to an HIV visit was 50% of the cost of a separate, non-integrated hypertension visit.

We estimate cost of treatment for acute stroke or myocardial infarction, when accessed, derived from public sector costs based on available data in the literature, adjusted for inflation

to 2024 US dollars [40]. Up to 80% of avoidable deaths due to CVD are due to poor quality care, with the remaining due to non-utilization of health systems [37], thus we assume that some acute care is ineffective. We further assume that ineffective care is less expensive than the cost of effective care due to more limited services or accessing care at a lower-level facility that lacks appropriate services but also at a lower cost (median 50%, range 25% to 100%; Table I, pages 21–23 of S1 Appendix). We explore a wide range of parameters for emergency care seeking (20% to 80% seeking care for acute moderate/severe CVD event) and effectiveness of such care for reducing acute risk of mortality (12.5% to 50%; Table D, pages 6–10 of S1 Appendix).

## Cost-effectiveness analysis

For each setting-scenario considered, we estimated the proportion of adults with hypertension who are diagnosed, on antihypertensive treatment, and had controlled hypertension over a 50-year time horizon beginning in 2024. We also estimated the absolute numbers of CVD events, costs, and DALYs for a population of 10 million adults. We use a 50-year time horizon to fully capture costs and benefits of hypertension treatment across the life course. We analysed costs and resource use from a health system perspective, summarizing costs based on location where services are delivered, namely "screening costs" as costs from community-based screening by CHWs, "clinic costs" as the costs associated with delivering hypertension care in clinics (personnel, supplies, facilities, clinician training), "drug costs" as the costs of the antihypertensive medications themselves, and "CVD care costs" as the costs of emergency CVD care provided in hospitals (not chronic CVD care).

Cost-effectiveness can be assessed by ranking policies in order of effectiveness, removing those that are less effective and more costly than an alternative (i.e., subject to dominance), and comparing the incremental cost-effectiveness ratio (ICER) or "Cost-per-DALY-averted" of each policy against the next most effective alternative [41]. We calcluated the ICER for each policy by dividing the incremental cost of the policy by the incremental DALYs averted by the policy compared to the next most effective policy [41]. We plot the incremental costs and incremental DALYs of CCC and CHW policies, compared to SOC for each of the 3,000 setting-scenarios generated by the model, showing their ICERs. The most effective policy which offers a cost-per-DALY-averted less than a cost-effectiveness threshold, which represents health systems ability to pay, is selected as being cost-effective. We used a cost-effectiveness threshold of $500 USD per DALY averted and a 3% discount rate for both costs and health outcomes to calculate DALYs averted and incremental net DALYs. Cost-effectiveness thresholds are uncertain; though a threshold of $500 USD which approximates the threshold estimated for lower-middle income countries (e.g., Kenya) and better accounts for opportunity costs than much higher previously used CETs of 1 to 3 times gross domestic product (GDP) [42–44]. In lower-income settings, a cost-effectiveness threshold of $500 per DALY averted presumes at least some degree of donor funding. We conducted sensitivity analysis to evaluate cost-effectiveness under varying cost-effectiveness thresholds to consider scenarios where funding is provided by domestic sources in lower income settings [42]. Additionally, we report the median and 90% range of net DALYs averted across this range of cost-effectiveness thresholds. To help understand budgetary impacts of each policy, we include a budget impact analysis to provide annual undiscounted cost estimates over the next 5 years. We also conducted an additional sensitivity analysis using a higher 5% discount rate for both costs and health outcomes, effectively reducing the impact of health effects and costs that occur in the more distant future on cost-effectiveness estimates. To further put findings in context, we provide a country profile of model estimates for Uganda (S1 Text).

We also calculated incremental net DALYs, a method for comparing cost-effectiveness between policies by estimating each policy's total health benefit and its opportunity costs [41]. Opportunity costs are defined by the cost-effectiveness threshold, representing the cost at which an averted DALY could be achieved through alternative health investments. Comparing policies against standard of care, this is calculated "Incremental Net DALYs = incremental DALYs + (incremental costs / cost-effectiveness threshold)." We report the proportion of setting-scenarios where each policy is cost-effective (i.e., incurrs the least net DALYs). To understand the impact of baseline setting-scenario characteristics, we also report the proportion of settings where each policy is the cost-effective choice among strata defined by selected baseline setting-scenario characteristics. We report our cost-effectiveness analysis in accordance with the 2022 Consolidated Health Economic Evaluation Reporting Standards (S1 CHEERS Checklist) [45].

## Results

### Hypertension care cascade

Over the next 50 years (2024 to 2074), the proportion of adults with hypertension aged 45 to 64 years who have received a hypertension diagnosis is expected to remain at 29% (90% range: 19% to 40%) under current conditions (Fig 2 and Table 3). Improving clinic-based care without added screening (CCC policy) would only improve the proportion diagnosed to 34% (21% to 48%), whereas additional CHW screening (CHW policy) would increase diagnosis to 79% (71% to 86%).

The proportion currently on antihypertensive treatment among adults aged 45 to 64 with hypertension would remain low at 6.7% (2.7% to 12%) under standard of care, whereas the proportion currently on treatment would increase to 19% (8.9% to 33%) with CCC implementation and 55% (45% to 64%) with CHW screening.

With expanded screening, there is risk of overdiagnosis and overtreatment. In the CHW policy, we estimate that 3.1% (2.3% to 3.9%) of normotensive adults would receive a hypertension diagnosis and 1.3% (0.8% to 1.9%) would be on treatment, with prevalence increasing with age (8.5% (6.4% to 11%) overdiagnosed and 3.6% (2.2% to 5.3%) overtreated among age ≥65). Estimated mean pre-treatment SBP among overdiagnosed individuals is 135 (135 to 136), indicating that this phenomenon is expected to occur mostly among those near the treatment threshold (SBP ≥140 mmHg) and likely to benefit from treatment.

We project that hypertension control among those with true hypertension (current or pre-treatment SBP ≥140 mmHg) would remain low at 3.6% (1.2% to 7.4%) among adults aged 45 to 64 years under current standard hypertension treatment conditions. Implementing the CCC policy alone would increase the proportion of those with hypertension maintaining blood pressure control to 14% (5.8% to 26%) and CHW screening to 44% (35% to 54%).

### Cardiovascular disease outcomes

Incident IHD events are projected to be 0.33 (0.17 to 0.59) per 100 person-years in all adults aged 45 to 64 under standard of care conditions with expected relative risk of 0.89 (0.79 to 0.98) with CCC implementation and 0.66 (0.55 to 0.79) with CHW screening. Stroke incidence is projected to be 0.33 (0.13 to 0.67) per 100 person-years in adults aged 45 to 64 under standard of care conditions with expected relative risk of 0.85 (0.73 to 0.96) with CCC implementation and 0.56 (0.43 to 0.7) with CHW screening. Projected relative risk for CVD mortality among all adults aged 18 and above is 0.91 (0.85 to 0.97) with CCC and 0.75 (0.66 to 0.83) with CHW policies. CCC implementation alone is expected to avert a mean of 19.8 (−2.5 to 45.7) thousand DALYs per year for a population of 10 million persons aged 15 and older, whereas

## A. Age ≥18 years

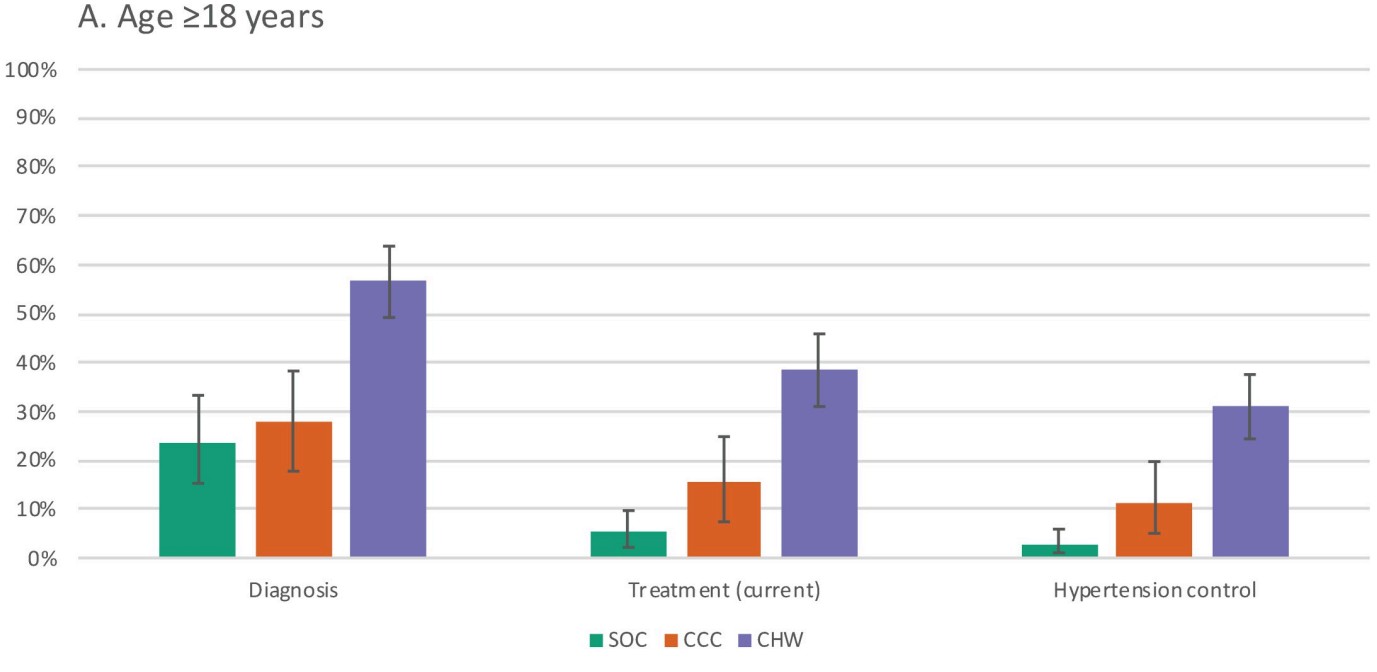

## B. Age 45-64 years

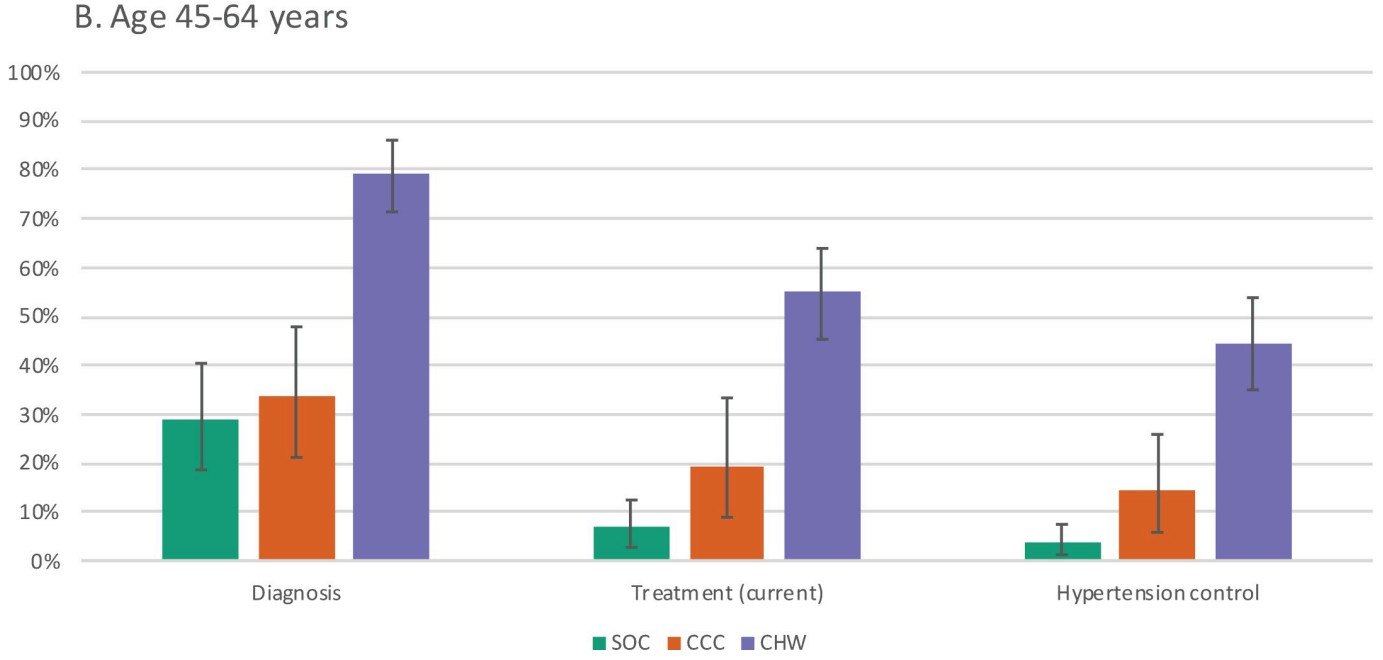

**Fig 2. Hypertension care cascade (2024–2074).** Mean proportion of hypertension diagnosis, current treatment, and control across setting-scenarios (90% range depicted by error bars) among adults aged ≥18 years (A) and 45–64 years (B). SOC, current standard of care; CCC, chronic care clinic; CHW, community health worker population-level screening (age ≥40).

the projected health benefits of CHW screening are more than 3-fold greater (61.5 (19.5 to 110.8) thousand DALYs averted annually).

**Table 3. Hypertension care cascade and cardiovascular disease outcomes of hypertension policy scenarios over 50 years (2024–2074); Mean (90% range).**

| | Age | SOC | CCC | CHW |
|---|---|---|---|---|
| Mean SBP | 25–44 | 125 (120 to 129) | 124 (120 to 129) | 124 (120 to 128) |
| | 45–64 | 137 (128 to 146) | 135 (127 to 144) | 130 (124 to 136) |
| | ≥65 | 143 (133 to 152) | 140 (131 to 148) | 134 (127 to 140) |
| Hypertension prevalence* | ≥18 | 29% (20% to 38%) | 29% (20% to 39%) | 30% (20% to 39%) |
| | 25–44 | 22% (15% to 29%) | 22% (15% to 29%) | 22% (15% to 29%) |
| | 45–64 | 44% (28% to 60%) | 45% (28% to 60%) | 44% (27% to 60%) |
| | ≥65 | 56% (38% to 70%) | 57% (39% to 71%) | 58% (39% to 73%) |
| Diagnosis | ≥18 | 23% (15% to 33%) | 28% (18% to 38%) | 57% (49% to 64%) |
| | 25–44 | 13% (7.7% to 19%) | 15% (8.6% to 22%) | 25% (19% to 31%) |
| | 45–64 | 29% (19% to 40%) | 34% (21% to 48%) | 79% (71% to 86%) |
| | ≥65 | 42% (29% to 57%) | 49% (33% to 65%) | 86% (79% to 92%) |
| Overdiagnosis† | ≥18 | 0.4% (0.2% to 0.7%) | 0.7% (0.3% to 1.3%) | 3.1% (2.3% to 3.9%) |
| | 25–44 | 0.3% (0.1% to 0.4%) | 0.4% (0.2% to 0.7%) | 0.8% (0.6% to 1.1%) |
| | 45–64 | 0.8% (0.4% to 1.3%) | 1.5% (0.6% to 3.2%) | 8.5% (6.4% to 11%) |
| | ≥65 | 1.5% (0.8% to 2.5%) | 3.2% (1.2% to 6.6%) | 15% (11% to 19%) |
| Current treatment | ≥18 | 5.3% (2.2% to 9.6%) | 15% (7.4% to 25%) | 39% (31% to 46%) |
| | 25–44 | 2.1% (0.8% to 4%) | 6.5% (2.8% to 12%) | 13% (8.9% to 18%) |
| | 45–64 | 6.7% (2.7% to 12%) | 19% (8.9% to 33%) | 55% (45% to 64%) |
| | ≥65 | 10% (4.3% to 19%) | 29% (14% to 46%) | 63% (54% to 72%) |
| Ever treatment | ≥18 | 20% (12% to 29%) | 26% (16% to 36%) | 53% (46% to 60%) |
| | 25–44 | 9.2% (5.1% to 15%) | 13% (7.2% to 19%) | 21% (16% to 27%) |
| | 45–64 | 24% (14% to 35%) | 32% (19% to 46%) | 74% (66% to 82%) |
| | ≥65 | 38% (24% to 53%) | 47% (31% to 64%) | 83% (76% to 90%) |
| Over-treatment (proportion of non-hypertensive individuals on treatment)‡ | ≥18 | 0% (0% to 0.1%) | 0.3% (0.1% to 0.7%) | 1.3% (0.8% to 1.9%) |
| | 25–44 | 0% (0% to 0.1%) | 0.1% (0% to 0.3%) | 0.3% (0.2% to 0.5%) |
| | 45–64 | 0.1% (0% to 0.2%) | 0.8% (0.2% to 1.9%) | 3.6% (2.2% to 5.3%) |
| | ≥65 | 0.2% (0.1% to 0.4%) | 1.7% (0.4% to 4.3%) | 6.5% (3.9% to 10%) |
| Hypertension control | ≥18 | 2.9% (1% to 5.8%) | 11% (5% to 20%) | 31% (24% to 37%) |
| | 25–44 | 1.4% (0.5% to 2.8%) | 5.4% (2.2% to 9.9%) | 10% (6.8% to 15%) |
| | 45–64 | 3.6% (1.2% to 7.4%) | 14% (5.8% to 26%) | 44% (35% to 54%) |
| | ≥65 | 5.4% (1.8% to 11%) | 21% (9.3% to 35%) | 49% (40% to 59%) |
| IHD incidence (per 100 person-years)§ | ≥18 | 0.25 (0.14 to 0.4) | 0.23 (0.13 to 0.36) | 0.18 (0.11 to 0.28) |
| | 25–44 | 0.04 (0.03 to 0.06) | 0.04 (0.03 to 0.06) | 0.04 (0.03 to 0.05) |
| | 45–64 | 0.33 (0.17 to 0.59) | 0.29 (0.15 to 0.49) | 0.21 (0.13 to 0.34) |
| | ≥65 | 1.63 (0.93 to 2.51) | 1.43 (0.84 to 2.18) | 1.1 (0.69 to 1.62) |
| Relative reduction in IHD incidence§ | ≥18 | NA | 0.9 (0.83 to 0.97) | 0.72 (0.64 to 0.81) |
| | 25–44 | NA | 0.98 (0.86 to 1.11) | 0.94 (0.82 to 1.06) |
| | 45–64 | NA | 0.89 (0.79 to 0.98) | 0.66 (0.55 to 0.79) |
| | ≥65 | NA | 0.88 (0.79 to 0.96) | 0.69 (0.6 to 0.79) |
| CVA incidence (per 100 person-years)§ | ≥18 | 0.26 (0.13 to 0.44) | 0.23 (0.11 to 0.38) | 0.16 (0.09 to 0.26) |
| | 25–44 | 0.02 (0.01 to 0.03) | 0.02 (0.01 to 0.03) | 0.02 (0.01 to 0.03) |
| | 45–64 | 0.33 (0.13 to 0.67) | 0.28 (0.12 to 0.52) | 0.18 (0.09 to 0.31) |
| | ≥65 | 1.87 (0.93 to 3.03) | 1.58 (0.85 to 2.51) | 1.14 (0.67 to 1.75) |
| Relative reduction in CVA incidence§ | ≥18 | NA | 0.87 (0.77 to 0.95) | 0.64 (0.53 to 0.75) |
| | 25–44 | NA | 0.97 (0.8 to 1.15) | 0.9 (0.74 to 1.08) |
| | 45–64 | NA | 0.85 (0.73 to 0.96) | 0.56 (0.43 to 0.7) |
| | ≥65 | NA | 0.85 (0.74 to 0.94) | 0.62 (0.52 to 0.74) |

(*Continued*)

**Table 3.** (Continued)

| | Age | SOC | CCC | CHW |
|---|---|---|---|---|
| CVD mortality rate (per 100 person-years) | ≥18 | 0.44 (0.27 to 0.63) | 0.4 (0.24 to 0.58) | 0.32 (0.21 to 0.45) |
| | 25–44 | 0.06 (0.05 to 0.09) | 0.06 (0.04 to 0.08) | 0.06 (0.04 to 0.08) |
| | 45–64 | 0.59 (0.34 to 0.94) | 0.53 (0.31 to 0.83) | 0.4 (0.26 to 0.58) |
| | ≥65 | 2.78 (1.85 to 3.83) | 2.47 (1.65 to 3.44) | 1.96 (1.37 to 2.62) |
| Relative reduction of CVD mortality | ≥18 | NA | 0.91 (0.85 to 0.97) | 0.75 (0.66 to 0.83) |
| | 25–44 | NA | 0.98 (0.89 to 1.08) | 0.96 (0.87 to 1.06) |
| | 45–64 | NA | 0.9 (0.81 to 0.97) | 0.69 (0.57 to 0.81) |
| | ≥65 | NA | 0.89 (0.81 to 0.96) | 0.71 (0.63 to 0.81) |
| CVD mortality rate (people with HIV; per 100 person-years) | ≥18 | 1.14 (0.51 to 2.33) | 0.83 (0.39 to 1.63) | 0.77 (0.36 to 1.5) |
| Relative risk of CVD mortality (people with HIV) | ≥18 | NA | 0.74 (0.57 to 0.9) | 0.69 (0.53 to 0.85) |
| All-cause mortality rate (per 100 person-years) | ≥18 | 1.36 (1.12 to 1.58) | 1.34 (1.11 to 1.56) | 1.29 (1.07 to 1.49) |
| Relative risk of all-cause mortality | ≥18 | NA | 0.98 (0.97 to 1) | 0.95 (0.92 to 0.98) |
| All-cause mortality rate (people with HIV; per 100 person-years) | ≥18 | 4.74 (3.08 to 8.11) | 4.56 (2.98 to 7.77) | 4.52 (2.97 to 7.67) |
| Relative risk reduction of all-cause mortality (people with HIV) | ≥18 | NA | 0.97 (0.84 to 1.09) | 0.96 (0.83 to 1.08) |

SOC, current standard of care; CCC, chronic care clinic; CHW, community health worker population-level screening (age ≥40); NA, not applicable. Data shown as means/proportions with 90% ranges.

* True underlying current or pre-treatment SBP ≥140 mmHg (regardless of whether measured).

† Overdiagnosis indicates the proportion of normotensive individuals erroneously given a diagnosis of hypertension.

‡ Overtreatment indicates the proportion of normotensive individuals diagnosed and treated for hypertension.

§ Moderate/severe events only. Mild events assumed to be asymptomatic and therefore under-ascertained in existing data from the literature used for model calibration.

CVD, cardiovascular disease; IHD, ischemic heart disease; SBP, systolic blood pressure.

## Costs and cost-effectiveness

Mean incremental costs compared to standard care, discounted over the 50 year time horizon, are $1.4 (−3.2 to 4.8) million USD for chronic care clinic implementation and $17.6 (5.4 to 28) million USD for additional CHW screening, scaled to a total population size of 10 million adults. The ICER was $69/DALY averted for CCC and $389/DALY averted for CHW policies (Fig 3). Using a cost-effectiveness threshold of $500, the CHW policy was cost-effective in 62% of scenarios, the CCC policy in 31%, and neither policy was cost-effective in 6.6% of scenarios (Table 4).

Stratifying on baseline setting-scenario characteristics (Table 5), community-wide CHW screening was more often cost-effective in settings with a greater baseline burden of hypertension and CVD mortality. Cost-effectiveness of the clinic-based CCC policy increased with greater HIV prevalence, though CHW screening remained the cost-effective strategy even in settings with the highest HIV prevalence. Cost-effectiveness of CHW screening increased with greater acute care utilization for CVD events (and thus higher acute care costs), though remained cost-effective in the majority of setting-scenarios (53%) where only 20% of people sought emergency care following an acute CVD event.

We also evaluated several alternative cost scenarios (Table 5). CHW screening (with CCC) was most sensitive to higher-cost scenarios. CHW screening was cost-effective in 35% of setting-scenarios where drug costs were doubled, indicating settings where high-quality drugs could not be obtained at generic prices. CHW screening were cost-effective in only 27% of setting-scenarios where clinical costs were doubled, indicating settings with higher opportunity costs of additional clinic visits. ICERs for alternative cost scenarios can be found in S1 Appendix (Section 5, pages 55 and 56).

## Cost–effectiveness frontier

Per 10 million adults over 50–year time horizon (2024 – 2074), n=3000 setting–scenarios

**Fig 3. Cost-effectiveness of hypertension treatment policies.** Costs in 2024 US Dollars. Future costs and DALYs averted were discounted by 3% per annum. CCC, chronic care clinic; CHW, community health worker; DALY, disability-adjusted life-year; SOC, standard of care.

In sensitivity analyses, we evaluate cost-effectiveness of each policy under varying cost-effectiveness thresholds, reporting the proportion of setting-scenarios where each policy is cost-effective as defined by incurring the fewest net DALYs at a given cost-effectiveness threshold (Fig 4A). When the presumed cost-effectiveness threshold is decreased to $300 USD per DALY averted, the CCC policy without community screening was cost-effective in 62% of modeled scenarios. When the cost-effectiveness threshold is further decreased to $100 USD per DALY averted, a threshold that is likely affordable even with domestic funding in lower-income countries in Africa, CCC implementation was the cost-effective policy in 50% of settings. Fig 4B demonstrates the median and 90% range of net DALYs averted across cost-effectiveness thresholds. It is apparent from this figure that for very low cost-effectiveness thresholds approaching $100 USD/DALY averted, the CHW policy is associated with a large negative number of net DALYs averted—suggesting large opportunity costs for implementing the CHW policy in settings with a very low cost-effectiveness threshold.

To aid policymakers, we present impacts on the hypertension care cascade and undiscounted annual costs over the first 5 years of implementation in S1 Appendix (Section 6, page 57), noting that undiscounted costs over the first 5 years are similar in magnitude to discounted costs presented in Table 4. In our sample country profile that is scaled to the population size of Uganda, implementation of the CHW policy is expected to avert approximately

**Table 4. Cost and cost-effectiveness of hypertension policies per 10 million adults over 50 years (2024–2074); Mean (90% range).**

| | SOC | CCC | CHW |
|---|---|---|---|
| DALYs Averted (thousands) | NA | 19.8 (−2.5 to 45.7) | 61.5 (19.5 to 110.8) |
| Costs (millions USD) | | | |
| Screening | NA | NA | 6.6 (5.8 to 7.6) |
| Clinic | 4 (1.5 to 7.7) | 5.4 (2.4 to 9.6) | 14.1 (7.9 to 21.2) |
| Drug | 1.1 (0.4 to 2.5) | 4 (1.6 to 7.3) | 10.2 (5.9 to 14.8) |
| CVD care | 23 (5.5 to 58) | 20.1 (4.9 to 50.6) | 14.8 (3.6 to 36.9) |
| Total HTN cost | 28.1 (9 to 64) | 29.5 (11.6 to 61.2) | 45.7 (26.8 to 72.3) |
| Incremental cost | NA | 1.4 (−3.2 to 4.8) | 17.6 (5.4 to 28) |
| ICER ($USD / DALY averted)* | NA | $69 | $389 |
| Incremental net DALYs averted (thousands)† | NA | 17 (−5.7 to 44.4) | 26.3 (−12.6 to 77.6) |
| Proportion of setting-scenarios where strategy is cost-effective (considering all 3 policies)† | 7% | 31% | 62% |

Scaled to population size of 10 million people aged ≥15 years.

SOC, current standard of care; CCC, chronic care clinic; CHW, community health worker population-level screening (age ≥40); CVD, cardiovascular disease; DALY, disability-adjusted life-year; ICER, incremental cost-effectiveness ratio. NA = not applicable. Data shown as means with 90% ranges. Population estimates among all adults aged ≥15 years.

\* For the CHW policy, the ICER is calculated as the incremental cost-effectiveness of adding CHW screening to CCC (the next most effective policy in terms of health benefit).

† Assuming cost-effectiveness threshold of $500 USD per DALY averted. Incremental net DALYs calculated by summing total average annual DALYs and average annual cost / cost effectiveness threshold of $500 USD, then comparing net DALYs averted to standard of care for each policy.

25,000 heart attacks and 34,000 strokes annually over the next 50 years, accounting for anticipated population growth and aging (S1 Text).

We conducted an additional sensitivity analysis of the CHW policy should CHW screening be limited to adults aged ≥50 years (full results in Section 7.1, pages 58–60 of S1 Appendix). Across 1,000 setting-scenarios, a more targeted screening approach limited to adults ≥50 years would lead to lower hypertension control among all adults (23%, 90% range 17% to 30%) and attenuated relative risk of CVD mortality (0.78, 0.70 to 0.85), but at lower incremental costs. The CHW screening policy restricting screening to adults age ≥50 was cost-effective in 72% of setting-scenarios.

In additional sensitivity analysis with a 5% discount rate for both costs and health benefits, the ICERs increased slightly for both CCC and CHW policies, to $77 and $444 USD/DALY averted. The CCC policy was cost-effective in 40% of setting-scenarios, the CHW policy in 51%, and neither policy was cost-effective in 9% of setting-scenarios (Section 7.2, pages 61 and 62 of S1 Appendix).

**Table 5.  Proportion of setting-scenarios where each hypertension policy is cost-effective over next 50 years (2024–2074), assuming a cost-effectiveness threshold of $500 USD per DALY averted.**

| Setting scenario characteristics in 2023 | | SOC | CCC | CHW |
|---|---|---|---|---|
| Mean SBP among adults aged 45–64 | <132 | 20% | 58% | 22% |
| | 132 to <136 | 9% | 48% | 43% |
| | 136 to <140 | 3% | 27% | 70% |
| | > = 140 | 0% | 7% | 93% |
| Hypertension prevalence among adults aged 45–64 | <35% | 20% | 53% | 27% |
| | 35 to <43% | 7% | 47% | 47% |
| | 43 to <50% | 4% | 25% | 71% |
| | > = 50% | 0% | 9% | 91% |
| Hypertension diagnosis among adults aged 45–64 | <22% | 7% | 21% | 72% |
| | 22 to <28% | 5% | 28% | 67% |
| | 28 to <36% | 8% | 32% | 60% |
| | > = 36% | 6% | 44% | 50% |
| Hypertension treatment among adults aged 45–64 | <4% | 8% | 23% | 69% |
| | 4 to <6% | 7% | 27% | 66% |
| | 6 to <9% | 7% | 34% | 59% |
| | > = 9% | 5% | 40% | 55% |
| Hypertension control among adults aged 45–64 | <2% | 5% | 18% | 77% |
| | 2 to <4% | 7% | 27% | 66% |
| | 4 to <6% | 8% | 38% | 55% |
| | > = 6% | 8% | 51% | 41% |
| CVD mortality rate among adults aged 45–64 | <400 per 100,000 | 17% | 56% | 28% |
| | 400 to <500 per 100,000 | 9% | 37% | 54% |
| | 500 to <600 per 100,000 | 3% | 33% | 64% |
| | > = 600 per 100,000 | 1% | 11% | 88% |
| Population-level mean SBP rise per decade after 2015 | 0 mmHg | 6% | 30% | 64% |
| | 0.5 mmHg | 8% | 33% | 60% |
| | 1 mmHg | 6% | 31% | 63% |
| Proportion of acute CVD events receiving emergency care | 20% | 9% | 37% | 53% |
| | 40% | 6% | 32% | 62% |
| | 80% | 4% | 24% | 71% |
| Proportion of emergency CVD care that is effective* | 12.5% | 7% | 33% | 59% |
| | 25% | 7% | 30% | 63% |
| | 50% | 6% | 30% | 64% |
| Cost of ineffective emergency CVD care compared to effective care | 25% | 8% | 35% | 56% |
| | 50% | 6% | 31% | 63% |
| | 100% | 6% | 27% | 67% |
| HIV prevalence among adults aged 15–49 | <5% | 8% | 24% | 68% |
| | 5 to <10% | 7% | 30% | 63% |
| | 10 to <15% | 5% | 36% | 59% |
| | > = 15% | 6% | 39% | 55% |

(*Continued*)

**Table 5.** (Continued)

| Setting scenario characteristics in 2023 | | SOC | CCC | CHW |
|---|---|---|---|---|
| Cost scenarios | Total costs 50% of base | 1% | 9% | 89% |
| | Acute CVD costs 25% of base | 11% | 43% | 46% |
| | Acute CVD costs 50% of base | 9% | 39% | 52% |
| | Drug costs 50% of base | 4% | 23% | 73% |
| | Clinic costs 50% of base | 4% | 20% | 77% |
| | Screening costs 50% of base | 5% | 20% | 76% |
| | Base case cost assumptions | 7% | 31% | 62% |
| | Screening costs 200% of base | 10% | 54% | 36% |
| | Clinic costs 200% of base | 15% | 59% | 27% |
| | Drug costs 200% of base | 16% | 49% | 35% |
| | Acute CVD costs 200% of base | 4% | 23% | 74% |
| | Acute CVD costs 400% of base | 2% | 12% | 87% |
| | Total costs 200% of base | 19% | 68% | 13% |

Table depicts the proportion of settings where a given policy is the most cost-effective (greatest number of net DALYs averted [DALYs averted–incremental costs / cost-effectiveness threshold], using a cost-effectiveness threshold of $500 USD/DALY averted). Row percentages total to 100%.

SOC, current standard of care; CCC, chronic care clinic; CHW, community health worker population-level screening (age ≥40).

* Effective care represents emergency care that includes timely delivery of evidence-based interventions that result in acute mortality risk reduction.

DALY, disability-adjusted life-year; SBP, systolic blood pressure.

## Discussion

Cardiovascular disease is a major public health problem in Africa and expected to grow in severity over the coming years [2]. In our modeling study, we demonstrate that population-wide hypertension screening by community health workers, combined with hypertension treatment implemented within integrated chronic care clinics that leverage existing HIV primary care infrastructure, would lead to large reductions in cardiovascular disease and is likely cost-effective in most settings. Even at lower cost-effectiveness thresholds likely to be affordable without donor funding in low-income countries, implementation of hypertension treatment within chronic care clinics at the primary health system level remained cost-effective in half of modeled settings compared to current standard of care where rates of hypertension diagnosis, treatment, and control are far too low [7].

A cornerstone of both strategies explored in this model is implementation of clinic-based hypertension care that is patient-centered, friendly, and built upon the existing HIV primary care system. This approach is informed by the hypertension care model in the SEARCH study where we have previously shown that patient-centered, clinic-based hypertension care can improve hypertension control and improve all-cause mortality in rural Kenya and Uganda [14]. The recent INTE-AFRICA study in Tanzania and Uganda also showed improvements in hypertension treatment outcomes with an integrated multi-disease approach to HIV, diabetes, and hypertension management [16], and other studies have also shown improved outcomes with integrated cardiovascular multimorbidity care models [46]. Key implementation strategies also include use of standardized treatment guidelines such as those developed by WHO HEARTS and Resolve To Save Lives [47,48]. Our modeling study demonstrates that such clinic-based approaches can improve outcomes for those already accessing care (largely persons with HIV) and may be highly cost-effective compared to current hypertension care by leveraging efficiencies of existing HIV primary care infrastructure. Expansion of hypertension

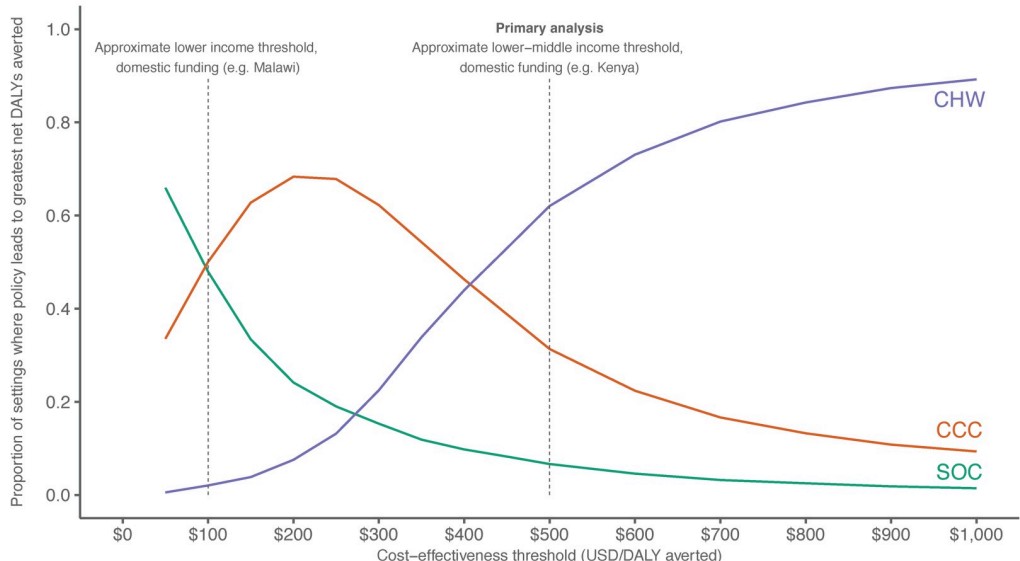

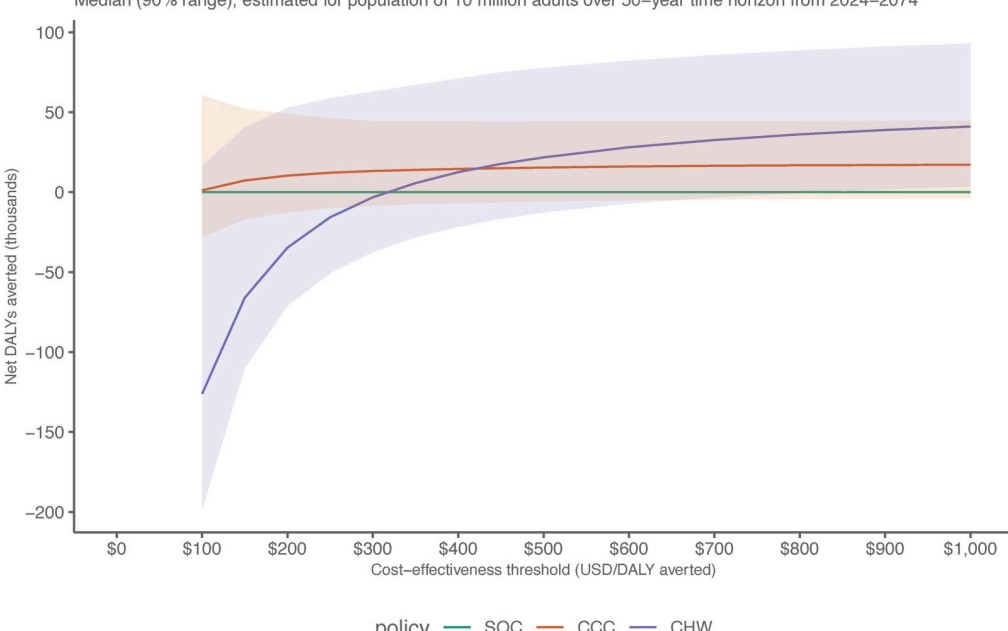

**Fig 4. Cost-effectiveness of hypertension policies at varying cost-effectiveness thresholds.** Panel A shows the proportion of setting-scenarios where a given policy is cost-effective (greatest number of net DALYs averted) across cost-effectiveness thresholds. Panel B shows the median (90% range) net DALYs averted for each policy plotted against varying cost-effectiveness thresholds. SOC, current standard of care; CCC, chronic care clinic; CHW, community health worker population-level screening (age ≥40).

services within primary health systems in Africa in a way that leverages the highly successful HIV platform is an important step towards universal health coverage [49].

However, in the absence of community-based screening, clinic-based chronic care implementation is not expected to meaningfully increase the number of people accessing hypertension treatment beyond those already receiving chronic disease care. There is extensive literature on the feasibility and effectiveness of hypertension screening by CHWs [17–22], and presence of an existing cadre of CHWs within many settings offers an opportunity for population-level reach of hypertension screening [23]. In our model, implementation of clinic-based chronic hypertension care alone is expected to reduce CVD-specific mortality by 26% among people living with HIV (who largely already routinely access clinic-based care) versus only 9% among the entire population. In contrast, adding population-level CHW screening for adults ≥40 years of age would reduce CVD-specific mortality by 25% among the entire adult population. Though CHW screening was more costly, this policy remained cost-effective at thresholds commonly applied to HIV by major global health funders in the majority of settings evaluated. CHW screening was, however, more sensitive to higher cost scenarios evaluated in sensitivity analysis, highlighting the need for strategies to reduce CHW screening costs. Our model assumed stand-alone screening for hypertension by CHWs, which may have increased screening cost estimates. One route to cost reduction includes integrated screening for hypertension alongside other conditions, such as HIV [50].

Regarding CHW compensation, our model assumes that CHWs are paid for their work, consistent with WHO guidelines [51]. This represents a change from current conditions where most CHWs are either volunteers or are inadequately renumerated [52,53]. We modeled CHW compensation based on the median minimum wage across countries in Africa with recently revised minimum wage policies; sensitivity analyses encompass the range of minimum wages across settings (Table J, page 24 of S1 Appendix). Addressing payment models, including cost-sharing across funding sources for multi-disease interventions and developing legal frameworks to protect CHW labor rights are important considerations for scale-up of CHW screening programs [52,53].

Our model assumes drugs are purchased by health systems at generic costs, available free of charge to patients, and of high quality. Patient-facing drug costs are a major barrier to retention in care and are a frequent source of catastrophic expenditures [40]. Drug quality issues are a challenge throughout low- and middle-income countries and undermine biologic effectiveness of hypertension treatment, patient trust, and provider assessment of adherence and treatment effects [54]. In this analysis, we assume that a portion of supply-chain costs will be directed toward quality monitoring and that procured drugs would be of high quality. Regarding drug costs, first-line antihypertensive drugs are inexpensive and in many settings can be obtained for less than $5 USD for a 1-year supply [39]. Additional cost-lowering through economies of scale would further increase cost-effectiveness in our analysis. Thus, global efforts to secure low-cost, high-quality antihypertensive medication supplies is an important priority. We also assumed that single drugs were used. Additional benefit may be derived if fixed-dose combination medications could be obtained at similar prices, given evidence for improved effectiveness of such formulations [55].

There are 2 clear policy implications of these findings. First, integration of hypertension care with existing HIV primary care systems can improve hypertension treatment outcomes and can be done cost-effectively in many settings. Second, global health funding agencies could have a large impact on the burden of CVD in Africa over the next several decades by raising additional funds to invest in community-based hypertension screening, which we found to be cost-effective but at a higher threshold that presumes external funding in many settings due to the absolute increases in costs that we estimate would be required to implement population-level CHW screening [42]. Though screening all adults ≥40 years of age would have greater health impact and is cost-effective in most settings based on cost-effectiveness

thresholds that presume donor funding, this policy may not be affordable in all settings. Screening focused on adults ≥50 or older would be less costly and could be considered for initial implementation. Implementation of these recommendations would be an important step towards integration of other chronic disease services and are consistent with global priorities to expand primary care services on the path to universal health coverage [49]. Initiatives such as Resolve to Save Lives are an important start [56], though larger investments by multi-lateral funding initiatives will be needed if we are to make headway towards addressing the rapidly growing burden of CVD in Africa [57].

Though our model demonstrates that substantial health benefits can be achieved through population-level hypertension screening and improving clinic-based care delivery, the estimated hypertension control achieved through these interventions remains far too low. Achieving greater hypertension control that exceeds 80-80-80 hypertension treatment targets is achievable and would further improve CVD outcomes [58]. Community-based hypertension treatment may address some barriers to sustained hypertension treatment engagement, such as transportation to clinic [59]. Fixed-dose combination therapy may also improve hypertension control, reduce side effects, and simplify dose-intensification [55]. Such interventions should be explored in future modeling studies to understand potential health outcomes and cost-effectiveness at scale.

In addition to limitations already discussed, a further limitation is that there is little population-level data on CVD morbidity and mortality in Africa. We calibrated model output to the best available data, including population-level data where available, and incorporated a range of parameters in our model to account for uncertainty. We also only modeled the effect of hypertension on ischemic heart disease and stroke. While these outcomes account for approximately 75% of hypertension-associated mortality [32], hypertension is also associated with other clinical outcomes (e.g., hypertensive heart disease, chronic kidney disease). Exclusion of these other hypertension-associated outcomes may underestimate the effectiveness of hypertension treatment strategies.

While there is strong evidence that CHWs can effectively screen for hypertension, there is limited data on population-level uptake of CHW hypertension screening and linakge to care at scale. We based screening and linkage estimates on CHW hypertension screening among 15,000 persons across 8 communities in 2 countries based on comprehensive census data [27], though uptake may vary in other settings. Implementation of other aspects of these policies may also vary between policies and across setting-scenario. We therefore include a wide range of potential screening uptake in our modeling parameters to address uncertainty both in policy effects and implementation across settings.

Though we estimated costs using the best available data, costs may vary substantially by setting. We conducted sensitivity analyses to vary the cost of different intervention components, as well as acceptable cost-effectiveness thresholds, to estimate outcomes under a variety of conditions. Further, we took a health system approach to cost-effectiveness, though cost-savings may be greater when taking a whole-of-society approach that accounts for reduced transport time and time off work with expansion of hypertension care to primary health centers.

An additional limitation is that we were not able to include all potential policy alternatives in our analysis. We selected 2 policies for consideration based on available data in the literature to suggest that these approaches may have substantial impact on population-level CVD prevention. In the future, our model is well positioned to incorporate additional policies as new data arises regarding the most impactful and efficient approaches to CVD prevention. Finally, while we assume continuation of current policies in our "standard of care" comparator group, there is considerable uncertainty in how the relevant policy comparator will change over the next 50 years. We assume continuation of current conditions in order to fully compare added

costs and health benefits of the policies under consideration. Our results provide policy guidance on the potential relative effects of policy options for addressing hypertension over the next 50 years, rather than absolute predictions under any one policy. Nevertheless, in most cost-effectiveness analyses, there is uncertainty over future projections.

## Conclusions

In summary, we demonstrate that population-level hypertension screening by community health workers and implementation of hypertension treatment through integrated chronic care clinics could substantially reduce cardiovascular disease morbidity and mortality and is likely to be cost-effective in most settings across Africa. Importantly, these strategies can leverage existing healthcare infrastructure for chronic care delivery, including the HIV primary care system and existing cadres of community health workers.

## Supporting information

**S1 Appendix. Contains extended methods, detailed description of model parameters, comparison of model estimates to contemporary data, and supplemental results.**
(PDF)

**S1 Text. Example country profile (Uganda).** In this file, we sample from the 3000 model runs based on charactersitics at study baseline (end 2023) that are similar to national and subnational settings in Uganda. We provide estimates of health effects, cost, and cost-effectiveness, scaled to the population of Uganda. The purpose of this sample country profile is to illustrate how model findings can be used to identify expected outcomes for a particular setting.
(PDF)

**S1 CHEERS Checklist. CHEERS reporting guideline for economic evaluations of health interventions.**
(PDF)

## Author Contributions

**Conceptualization:** Matthew D. Hickey, James Ayieko, Jane Kabami, Asiphas Owaraganise, Elijah Kakande, Sabina Ogachi, Colette I. Aoko, Erick M. Wafula, Norton Sang, Helen Sunday, Paul Revill, Loveleen Bansi-Matharu, Starley B. Shade, Gabriel Chamie, Laura B. Balzer, Maya L. Petersen, Diane V. Havlir, Moses R. Kamya, Andrew N. Phillips.

**Data curation:** James Ayieko, Jane Kabami, Asiphas Owaraganise, Elijah Kakande, Sabina Ogachi, Colette I. Aoko, Erick M. Wafula, Norton Sang, Helen Sunday, Starley B. Shade.

**Formal analysis:** Matthew D. Hickey, Paul Revill, Andrew N. Phillips.

**Funding acquisition:** Matthew D. Hickey, Maya L. Petersen, Diane V. Havlir, Moses R. Kamya.

**Methodology:** Matthew D. Hickey, Paul Revill, Loveleen Bansi-Matharu, Andrew N. Phillips.

**Resources:** Andrew N. Phillips.

**Software:** Andrew N. Phillips.

**Supervision:** Moses R. Kamya, Andrew N. Phillips.

**Writing – original draft:** Matthew D. Hickey.

**Writing – review & editing:** James Ayieko, Jane Kabami, Asiphas Owaraganise, Elijah Kakande, Sabina Ogachi, Colette I. Aoko, Erick M. Wafula, Norton Sang, Helen Sunday, Paul Revill, Loveleen Bansi-Matharu, Starley B. Shade, Gabriel Chamie, Laura B. Balzer, Maya L. Petersen, Diane V. Havlir, Moses R. Kamya, Andrew N. Phillips.

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
