## [Editor Report · Decision Letter 0]

19 Aug 2024

Dear Dr Hickey, 

Thank you for submitting your manuscript entitled "Cost-effectiveness of leveraging existing HIV primary health systems and community health workers for hypertension screening and treatment in Africa: an individual-based modelling study" for consideration by PLOS Medicine.

Your manuscript has now been evaluated by the PLOS Medicine editorial staff as well as by an academic editor with relevant expertise and I am writing to let you know that we would like to send your submission out for external peer review.

Please re-submit your manuscript within two working days, i.e. by Aug 21 2024 11:59PM.

Kind regards,

Katrien G. Janin, PhD

Senior Editor

PLOS Medicine

---

## [Decision Letter · Decision Letter 1]

5 Nov 2024

Dear Dr Hickey,

Many thanks for submitting your manuscript "Cost-effectiveness of leveraging existing HIV primary health systems and community health workers for hypertension screening and treatment in Africa: an individual-based modelling study" (PMEDICINE-D-24-02699R1) to PLOS Medicine. The paper has been reviewed by subject experts and a statistician; their comments are included below and can also be accessed here: [LINK]

After discussing the paper with the editorial team and an academic editor with relevant expertise, I'm pleased to invite you to revise the paper in response to the reviewers' comments. We plan to send the revised paper to some or all of the original reviewers, and we cannot provide any guarantees at this stage regarding publication.

We ask that you submit your revision by Nov 26 2024 11:59PM. However, if this deadline is not feasible, please contact me by email, and we can discuss a suitable alternative.

Don't hesitate to contact me directly with any questions (lgaynor@plos.org). 

Best regards, 

Louise 

Louise Gaynor-Brook, MBBS PhD 

Senior Editor

PLOS Medicine

lgaynor@plos.org

Comments from the academic editor:

1. I am interested in the rationale for hypertension based screening over CVD risk based population screening based on WHO guidelines (e.g. https://iris.who.int/bitstream/handle/10665/333221/9789240001367-eng.pdf). A risk based strategy is more likely to identify appropriate people for treatment compared with a hypertension strategy - see this paper which suggests a both under- and over-treatment based on risk https://journals.plos.org/plosmedicine/article?id=10.1371/journal.pmed.1003485

2. The CHW intervention effects seem overly optimistic- increasing diagnosis rates from 29% to 79% in 45-64 years age group, control rates up to >40% and a RRR of 0.66 for CVD events. The authors state treatment effects are based on SEARCH study results. Could these assumptions be made more explicit in the methods? I note reviewer 3 also sought more information on the implementation model.

Comments from the reviewers: 

Reviewer #1: I congratulate the authors on the results of their hard work. This manuscript offers a valuable contribution by evaluating the cost-effectiveness of leveraging HIV health infrastructure and community health workers to tackle hypertension in Africa. The work is well-developed, and the supplemental material is extensive and relevant. I especially appreciate the detailed justification for all the parameter assumptions and the inclusion of the budget impact analysis results. It is also commendable that the manuscript has been drafted in accordance with the Consolidated Health Economic Evaluation Reporting Standards (CHEERS) guidelines.

There are just a few comments from my side:

-The phrase '3000 setting scenarios' was somewhat confusing. Upon checking the appendix, I found that what is meant appears to be more like '3000 Monte Carlo replications' for probabilistic sensitivity analysis (PSA), where parameters are derived from a given distribution simultaneously. Therefore, it may not be appropriate to call it '3000 setting scenarios'. Alternatively, further explanation could clarify this.

-The indexing of the images in the article seems to be incorrect. Please carefully check and revise them.

-Abbreviations should be added for 'Figure 1. Hypertension care cascade (2024-2074)'.

-In the methods section, it is stated that the costs were adjusted for inflation to 2023 US dollars, but Figure 2 presents costs in 2024 US dollars. Please ensure consistency.

Good luck!

Reviewer #2: Thank you to the authors for submitting this paper. It is clearly on a very important topic and I really enjoyed reading it. I have a few (relatively minor) comments that I think should be addressed before publication.

Methods

I know the authors have provided detailed methods and parameters in the appendix, however I would like some of this brought into the main text. The reality is that not many readers will venture into the appendices to look at the methods, and as this is (for me) the most important part of the paper I would like it to be expanded. Specific things I would bring into the main are a model schematic and table with some of the key variables and their sources.

Policy Comparison

The authors compare the intervention to two other policies. I was wondering if the authors could comment on whether there are any other relevant comparators that could have been included? I appreciate that given the limitations in the available data it may not have been possible to include them however an acknowledgment of any excluded relevant comparators would be useful (if applicable). For me this is especially relevant given the heterogeneity in the African countries included in the analysis. I note that the authors include a discussion of the fact that the most relevant comparator will change in the future years.

Cost Effectiveness Analysis

Could the authors please further justify the use of the "$500 per DALY averted" threshold in the base case? I note in Figure 3 this is referred to as the upper bound of the HIV threshold

Reviewer #3: In general this is a very thorough manuscript on the impact and cost-effectiveness of leveraging existing HIV-related healthcare infrastructure to decrease the burden of hypertension. I do, however, have a few significant points for consideration-

1. My biggest point of interest is around the 'how' of implementation and what these interventions precisely mean. The difference in these three interventions, I assume, ultimately come down to differences in uptake/care seeking if hypertension care is provided in different ways, and retention in care after the initial diagnosis. If this is the case, there needs to be a better description of what the SOC, CCC and CHW are and what they effectively mean in practice. What is the SOC, how do people typically seek care now, when would someone be offered to be screened? In practice for CCC, what would change, how would that mechanistically alter who receives care? A proper description in this main text of this manuscript without needing to refer to the SEARCH study is needed for all three scenarios.

2. For the CHW scenario- is this community testing? Door to door? Assumptions regarding the 'how' will alter effectiveness of the intervention and who is ultimately tested. If this is door to door, what is assumed about how many people a CHW can reach per day? What proportion are rural vs urban (in some settings). If it's door to door, what transport costs are being assumed (as community-level screening is typically extraordinarily costly due to things like transport and how spread-out people are in some settings, and the cost of general logistics and planning). In the supplemental material, I see that the cost estimates also come from SEARCH, but how 'transportable' are those estimates to a 'general' population across sub-Saharan Africa? (If costs of implementation varied greatly by things such as geography, population density, etc., then the costs should be weighted to reflect an 'average' population across the region) Also, what exactly was included in that $3 per person screened?

3. Are there any other costs that would need to be included when moving from SOC to CCC? Such as the cost of integrating information systems, renovations to clinics?

4. Within the SEARCH study, were there population- or implementation-specific factors that increased or decreased the uptake of initial diagnosis? Was any of that granularity used to refine the modelled results (e.g. were gender-specific estimates, or rural/urban-specific estimates used from the SEARCH study within the agent-based model?)?

5. Are there additional scenarios that can be explored, or other types of interventions that could be assessed that could increase uptake in care? (e.g. integrated care + demand generation, integrated care + targeted CHW outreach (say, in urban areas, or in areas with higher prevalence of hypertension), integrated care + mobile apps, screening integrated into the OPD, etc) Without exhausting all potential competing alternatives, it would be difficult to say whether the two interventions evaluated in this study are consistently on the cost-effectiveness frontier or not. 

Reviewer #4: Please see the attached review.

* Please include continuous line numbers in your revised version (i.e. not starting from one with each new page).

* Please upload any figures associated with your paper as individual TIF or EPS files with 300dpi resolution at resubmission; please read our figure guidelines for more information on our requirements: http://journals.plos.org/plosmedicine/s/figures. While revising your submission, please upload your figure files to the PACE digital diagnostic tool, https://pacev2.apexcovantage.com/. PACE helps ensure that figures meet PLOS requirements. To use PACE, you must first register as a user. Then, login and navigate to the UPLOAD tab, where you will find detailed instructions on how to use the tool. If you encounter any issues or have any questions when using PACE, please email us at PLOSMedicine@plos.org.

FIGURES AND TABLES

SUPPLEMENTARY MATERIAL

REFERENCES

MODELLING STUDIES

The following list is derived from Geoffrey P Garnett, Simon Cousens, Timothy B Hallett, Richard Steketee, Neff Walker. Mathematical models in the evaluation of health programmes. (2011) Lancet DOI:10.1016/S0140-6736(10)61505-X: 

* If pertinent, please provide a diagram that shows the model structure, including how the natural history of the disease is represented, the process and determinants of disease acquisition, and how the putative intervention could affect the system.

* Please provide a complete list of model parameters, including clear and precise descriptions of the meaning of each parameter, together with the values or ranges for each, with justification or the primary source cited and important caveats about the use of these values noted.

* Please provide a clear statement about how the model was fitted to the data, including goodness-of-fit measure, the numerical algorithm used, which parameter varied, constraints imposed on parameter values, and starting conditions.

* For uncertainty analyses, please state the sources of uncertainties quantified and not quantified [can include parameter, data, and model structure].

* Please provide sensitivity analyses to identify which parameter values are most important in the model. Uncertainty estimates seek to derive a range of credible results on the basis of an exploration of the range of reasonable parameter values. The choice of method should be presented and justified.

* Please discuss the scientific rationale for the choice of model structure and identify points where this choice could influence conclusions drawn. Please also describe the strength of the scientific basis underlying the key model assumptions.

* For studies that develop a prediction model or evaluate its performance, please ensure that the study is reported according to the TRIPOD statement (https://www.equator-network.org/reporting-guidelines/tripod-statement) and include the completed checklist as Supporting Information. Please add the following statement, or similar, to the Methods: "This study is reported as per the Transparent Reporting of a Multivariable Prediction Model for Individual Prognosis Or Diagnosis (TRIPOD) statement (S1 Checklist)." For studies using machine learning, please use the TRIPOD-AI checklist. When completing the checklist, please use section and paragraph numbers, rather than page numbers. 

HEALTH ECONOMICS / COST-EFFECTIVENESS STUDIES

* Please ensure that the study is reported according to the CHEERS guideline (available from: https://www.equator-network.org/reporting-guidelines/cheers) and include the completed checklist as Supporting Information. Please add the following statement, or similar, to the Methods: "This study is reported as per the Strengthening the Consolidated Health Economic Evaluation Reporting Standards 2022 (CHEERS 2022) Statement (S1 Checklist)." When completing the checklist, please use section and paragraph numbers, rather than page numbers.

---

## [Decision Letter · Decision Letter 2]

17 Dec 2024

Dear Dr. Hickey,

Thank you very much for re-submitting your manuscript "Cost-effectiveness of leveraging existing HIV primary health systems and community health workers for hypertension screening and treatment in Africa: an individual-based modelling study" (PMEDICINE-D-24-02699R2) for review by PLOS Medicine.

I have discussed the paper with my colleagues and it was also seen again by four reviewers. I am pleased to say that provided the remaining editorial and production issues are dealt with we are planning to accept the paper for publication in the journal.

[LINK]

We expect to receive your revised manuscript within the next few weeks. Please email us (plosmedicine@plos.org) if you have any questions or concerns.

We look forward to receiving the revised manuscript by Jan 03 2025 11:59PM.   

Sincerely,

Rebecca Kirk

On behalf of:

Louise Gaynor-Brook, MBBS PhD

Senior Editor 

PLOS Medicine

plosmedicine.org

Requests from Editors:

GENERAL EDITORIAL REQEUSTS

* At this stage, we ask that you include a short, non-technical Author Summary of your research to make findings accessible to a wide audience that includes both scientists and non-scientists. The Author Summary should immediately follow the Abstract in your revised manuscript. This text is subject to editorial change and should be distinct from the scientific abstract. Ideally each sub-heading should contain 2-3 single sentence, concise bullet points containing the most salient points from your study. In the final bullet point of ‘What Do These Findings Mean?’ Please include the main limitations of the study in non-technical language.

Please see our author guidelines for more information: https://journals.plos.org/plosmedicine/s/revising-your-manuscript#loc-author-summary.

* Please confirm that your title complies with to PLOS Medicine's style. Your title must be nondeclarative and not a question. It should begin with main concept if possible. "Effect of" should be used only if causality can be inferred, i.e., for an RCT. Please place the study design ("A randomized controlled trial," "A retrospective study," "A modelling study," etc.) in the subtitle (ie, after a colon).

* Please confirm that your abstract complies with our requirements, including providing all the information relevant to this study type https://journals.plos.org/plosmedicine/s/submission-guidelines#loc-abstract

* Please ensure that the Introduction ends with a clear description of the study question or hypothesis.

* Please ensure that all abbreviations are defined at first use throughout the text.

GENERAL

* Please provide the author contributions in full according to the format outlined for PLOS Medicine https://journals.plos.org/plosmedicine/s/authorship#loc-author-contributions

* Please adjust all references to the Appendix to be to the Supporting Information, and format accordingly https://journals.plos.org/plosmedicine/s/supporting-information

FUNDING STATEMENT

* The funding statement should include: specific grant numbers, initials of authors who received each award, URLs to sponsors’ websites. Also, please state whether any sponsors or funders (other than the named authors) played any role in study design, data collection and analysis, the decision to publish, or preparation of the manuscript. If they had no role in the research, include this sentence: “The funders had no role in study design, data collection and analysis, decision to publish, or preparation of the manuscript.”

COMPETING INTERESTS STATEMENT

* All authors must declare their relevant competing interests per the PLOS policy, which can be seen here: https://journals.plos.org/plosmedicine/s/competing-interests For authors with ties to industry, please indicate whether any of the interests has a financial stake in the results of the current study.

DATA AVAILABILITY

* Thank you for agreeing to make your data available. At this time, please provide the link to the data repository and accession numbers required for access.

FIGURES

* Please show graph axes beginning at zero. If this is not possible, please show a break in the axis.

ECONOMIC ANALYSES INCLUDING COST-EFFECTIVENESS ANALYSES

* Please report your economic analysis according to the appropriate study design provided at http://www.equator-network.org/?post_type=eq_guidelines&eq_guidelines_study_design=economic-evaluations&eq_guidelines_clinical_specialty=0&eq_guidelines_report_section=0&s= and provide the relevant completed checklist. In the checklist please include sufficient text excerpted from the manuscript to explain how you accomplished all applicable items.

Comments from Reviewers:

Reviewer #1: The effort the author has put into addressing the concerns and comments raised in the previous review is appreciated.

The revised manuscript has resolved the issues previously highlighted. The author's responses to the comments were clear and comprehensive, and the additional information provided has strengthened the study's methodology and conclusions. 

Overall, the manuscript is now well-structured and provides valuable insights into the topic. I have no additional substantive concerns. 

Good luck!

Reviewer #2: Thank you to the authors for integrating my comments into the paper. I'm happy to approve for publication. 

Reviewer #3: The authors have thoroughly responded to all of my comments and queries, and have done a commendable job making further improvements to their their manuscript.

Reviewer #4: Thank you for addressing the comments in my initial review.

[LINK]

---

## [Editor Report · Decision Letter 3]

10 Jan 2025

Dear Dr Hickey, 

On behalf of my colleagues and the Academic Editor, David Peiris, I am pleased to inform you that we have agreed to publish your manuscript "Cost-effectiveness of leveraging existing HIV primary health systems and community health workers for hypertension screening and treatment in Africa: an individual-based modelling study" (PMEDICINE-D-24-02699R3) in PLOS Medicine.

PRESS

Sincerely, 

Rebecca Kirk

On behalf of:

Louise Gaynor-Brook, MBBS PhD 

Senior Editor 

PLOS Medicine